# Bladder Oversensitivity Is Associated with Bladder Outlet Obstruction in Men

**DOI:** 10.3390/jpm12101675

**Published:** 2022-10-08

**Authors:** Guy Verhovsky, Ilia Baberashvili, Yishai H. Rappaport, Dorit E. Zilberman, Amos Neheman, Jonathan Gal, Amnon Zisman, Kobi Stav

**Affiliations:** 1Department of Urology, Shamir Medical Center (Assaf Harofeh Campus), Zerifin, Sackler School of Medicine, Tel Aviv University, Tel Aviv 6997801, Israel; 2Department of Nephrology, Shamir Medical Center (Assaf Harofeh Campus), Zerifin, Sackler School of Medicine, Tel Aviv University, Tel Aviv 6997801, Israel; 3Department of Urology, Samson Assuta Medical Center, Ashdod, Goldman School of Medicine, Ben-Gurion University of the Negev, Beer Sheva 84105, Israel; 4Department of Urology, Sheba Medical Center, Sackler School of Medicine, Tel Aviv University, Tel Aviv 6997801, Israel

**Keywords:** bladder outlet obstruction, urodynamics, bladder oversensitivity, overactive bladder

## Abstract

Objective: The aim of this study was to assess whether there is an objective association between bladder outlet obstruction (BOO) and abnormal sensation parameters during filling cystometry in men. Methods: This was a prospective study. Consecutive patients referred for urodynamic examination were assessed for eligibility. Patients with permanent catheters, BPH related surgery, neurologic disease, or inability to complete the urodynamic study were excluded. All patients underwent full physical examination, as well as renal and bladder ultrasound including prostate size estimation, post void residual volume, and PSA, and they completed the International Prostate Symptoms Score (IPSS) questionnaire. The cohort was divided into obstructed and un-obstructed groups according to the Bladder Outlet Obstruction Index. Results: Ninety of the 115 patients recruited were obstructed (78%). Obstructed patients had significantly higher PSA, larger prostate volume, and higher IPSS. Detrusor overactivity did not differ between the two groups (45.6% vs. 48.1%, *p* = 0.83). First, normal, strong, and urgent desires to void were significantly lower in obstructed men: median (IQR) 118 (57–128) vs. 180 (80–200), 171 (85–257) vs. 227 (125–350), 221 (150–383) vs. 307 (180–477), and 276 (197–480) vs. 344 (280–535), respectively. First desire to void (FDV) had the highest area under the curve (AUC = 0.83, 95% CI = 0.76–0.90, *p* < 0.001) for predicting BOO with a Youden index of 0.78 at 140 mL. Conclusions: Our results suggest that there is a strong association between bladder oversensitivity and BOO in men. Men with FDV <140 mL had a significantly increased probability of being obstructed. These findings may shed a light on the pathophysiological connection between obstruction and enhanced afferent signaling from the bladder.

## 1. Introduction

The prevalence of lower urinary tract symptoms (LUTS) has increased over the past 15 years, with nearly 63% of the worldwide population having experienced at least one adverse lower urinary tract symptom in 2018 [1]. Bladder outlet obstruction (BOO) and overactive bladder (OAB) each comprise approximately 21% of the reported LUTS etiologies, with a 57% overlap [1]. Twenty-seven percent of men with LUTS reporting OAB symptoms responded to alpha-blocker monotherapy with relief of their storage symptoms [2].

Post prostatectomy, one-third of patients demonstrated relief of OAB symptoms [1,3,4]. This phenomenon might be explained by morpho-functional alterations involving the detrusor muscle as a result of BOO’ experimental models have shown that bladder outlet obstruction causes detrusor smooth muscle cells hypertrophy and hyperplasia, as well as extracellular matrix alterations that may lead to bladder detrusor overactivity or increased bladder sensation (bladder oversensitivity) [2,3,4].

More than half of LUTS patients reporting characteristic OAB symptoms do not have detrusor overactivity (DO) upon urodynamic testing, and only half have detrusor overactivity that correlates with urge perception [5,6,7]. These phenomena are hypothesized to be two different phases (early and late) of the same pathological state; alternatively, they might represent two different subtypes of OAB [7]. Bladder oversensitivity, as defined by the International Continence Society (ICS), is an increased perceived bladder sensation during bladder filling with cystometric findings of early desire to void without an increase in detrusor pressure [8].

Recently, a strong association between bladder oversensitivity and BOO in women was reported [9]. In this study, we assessed the presumed association in the male population.

## 2. Materials and Methods

The study protocol was approved by the local ethical committee for experiments in humans, and all patients provided written informed consent (IRB 0101-17).

Male patients referred for a urodynamic study at our center were prospectively recruited. Exclusion criteria included an indwelling urethral catheter, known neurologic disease, bladder stone or tumor, pre-existing urethral pain (e.g., patients with chronic pelvic pain syndromes and bladder pain syndrome/interstitial cystitis), any prior prostate related surgery, and active urinary tract infection.

Comprehensive medical and urological histories were recorded. All patients underwent a physical examination, as well as renal and bladder ultrasound including prostate size estimation, post void residual volume, and PSA, and they completed the International Prostate Symptoms Score (IPSS) questionnaire [8]. The cohort was divided into obstructed and unobstructed groups, as defined by the Bladder Outlet Obstruction Index (BOOI = Pdet@Qmax − 2Qmax > 40 CmH_2_O).

All subjects ceased alpha blocker, anticholinergic, and beta 3 agonist medication 3 days prior to the urodynamic examination.

Subsequent to free uroflowmetry, the bladder was emptied using a 12F Tieman catheter. Multichannel urodynamic studies were performed using the AQUARIUS CT (Laborie Medical Technologies, Mississauga, ON, Canada) using dual-lumen 7F urethral and 7F rectal air-charged catheters. Filling was carried out using normal saline at body temperature with a filling rate of 50 mL/min. Filling and voiding cystometry was performed in supine and standing positions, respectively. 

All urodynamic studies were performed by a single senior urologist (K.S.). Bladder sensation volumes were assessed systematically according to International Continence Society Good Urodynamic Practices [10]: first desire to void (FDV), normal desire to void, strong desire, urgency, and maximal cytometric capacity (MCC). First sensation volume was not measured since it was shown to be an unreliable measure due to hypervigilance or overconcentration by the patient [11]. 

Participants with BOOI over 40 cmH_2_O were allocated to the obstructed group.

The sample size was calculated using WINPEPI 23.0; power was set to 80% with a *p*-value of 5%. To achieve an odds ratio of at least 1.5/0.7, a cohort of 70 men with outflow obstruction and 20 controls was calculated. 

All statistical analyses were performed using SPSS software, version 18.0 (IBM SPSS, Chicago, IL, USA). A *p*-value <0.05 was considered significant for all tests. Normally distributed data were expressed as the mean ± SD. Variables not following a normal distribution were expressed as medians and interquartile ranges (quartiles 1–3), and categorical variables were expressed as frequencies. Normally distributed continuous variables were compared between the two groups using a *t*-test. Variables with skewed distribution were compared using nonparametric Mann–Whitney U tests, and the chi-square test was used for comparison of categorical variables. The cutoffs for the most accurate discrimination of BOO risk for the sensation volumes were derived using standard receiver operating characteristic (ROC) curves. 

To further explore the association of bladder sensation volumes with BOO, univariate and multivariable logistic regression models were constructed. The confounders for multivariable models were chosen from univariate data analyses set with *p* < 0.25. Univariate and multivariate logistic regression analyses are presented as odds ratios (OR) with 95% confidence intervals (CI).

## 3. Results

In the first phase of the trial, 173 patients were assessed for eligibility. Twenty-eight were excluded and 30 were withdrawn due to either inability to pass the urinary catheter or due to inability to void with the urethral catheter in place. The flow of participants through each phase of the trial is presented in Figure 1. A total of 115 men (mean age 63 ± 14 years) completed the study; 90 (78%) were obstructed. 

The mean age of the obstructed and unobstructed patients was 57 ± 13 and 54 ± 12 years, *p* = 0.25, respectively. Study groups did not differ in prior use of alpha blockers or antimuscarinic medications: 15% vs. 16%, *p* = 0.66 and 3% vs. 4%, *p* = 0.52, respectively. Demographic, clinical, and urodynamic variables of both groups are presented in Table 1. Obstructed patients had higher PSA values [median 1.5 (range 0.7–3.2) vs. 0.5 (0.1–1.1), *p* = 0.005], larger prostate volume [median 35 (range 20–50) vs. 30 (20–35), *p* = 0.03], and higher IPSS scores [median 22 (range 16–29), vs. 17 (13–24), *p* = 0.01]. Median post void residual in the obstructed and unobstructed groups was 47 mL (IQR 16–68) vs. 38 mL (IQR 10–51), *p* = 0.38, respectively.

The detrusor overactivity (DO) rate was similar amongst the two groups (45.6% vs. 48.1%, *p* = 0.83). During the urodynamic filling phase, first, normal, strong, and urgent desire to void volumes were significantly lower in the obstructed group. Most significantly, first desire to void (FDV) among obstructed men was 118 mL (57–128), while, in unobstructed men, FDV was 180 mL (80–200), *p* < 0.0001 (Table 1).

Associations between bladder sensation volumes and BOO according to univariate and multivariate logistic regression analysis are described in Table 2. In univariate analysis, all the sensation volumes were significantly associated with BOO except for maximal cystometric capacity (MCC). In multivariate analysis, FDV was the only significant predictor for BOO, with an OR of 1.42 (1.25–1.68, *p* = 0.001) for every 50 mL decrease in bladder volume.

Of the sensation parameters, FDV had the highest area under the curve (AUC = 0.83, 95% CI = 0.76–0.90, *p* < 0.001) with a Youden index of 0.78 at 140 mL for predicting BOO (Figure 2). Thus, men with FDV <140 mL had the highest association with BOO.

## 4. Discussion

Bladder oversensitivity in the presence of BOO was previously demonstrated in women [9,12]. Bladder sensation volumes were significantly lower in obstructed women than non-obstructed women in univariate and multivariate logistic regression analysis. Similar to our study, first desire to void (FDV) had the highest area under the curve (AUC = 0.75, 95% CI = 0.672–0.837, *p* < 0.001) for predicting BOO. FDV <105 mL showed a strong association with BOO with OR = 9.84 (95% CI 4.122–23.508, *p* < 0.0001).

In this study, we demonstrated that an association between BOO and bladder oversensitivity exists in men as well.

Excluding MCC, all sensation volumes were significantly lower in men with BOO. In univariate and multivariate analyses, FDV had the strongest association with BOO. The prevalence of DO was similar between obstructed and non-obstructed men, emphasizing the significance of bladder oversensitivity.

In males, BOO (mostly due to benign prostatic enlargement) is the most common leading cause for OAB [3,13]. As the etiology of the obstruction is different from those of women, including the pressure changes in the bladder needed to void in both genders, and as the possible pathophysiologic changes to the bladder may differ, we believe that validating a similar trend in men is an important step for future studies looking at the clinical implementations of these findings.

Storage symptoms such as frequency and urgency in patients with BOO are widely considered to be associated with OAB. Lee et al. demonstrated urodynamically verified OAB in nearly half of the 144 consecutive male patients with BOO assessed. Approximately 30% of men with OAB improved on doxazosin, an alpha blocker [14]. 

Chen et al., in their review article, found BOO as one of the main etiologies for refractory overactive bladder. Of men with BOO and OAB, only one-third respond to alpha-blocker monotherapy, whereas combined therapy with an alpha-blocker and antimuscarinics improved LUTS in three-quarters of patients.

The pathophysiological explanation for OAB in BOO is still unclear. Animal experiments have shown that ischemia/reperfusion injury (I/RI) during BOO causes structure and function alterations in the lower urinary tract [6,15,16,17]. Gosling et al. found that partial ligation of the animal urethra causes detrusor muscle hypertrophy and, furthermore, triggers hypertrophy of afferent and efferent neurons. A reduction in contractile function and a stimulation of spontaneous electrical activity occur due to bladder outlet obstruction and detrusor mass enlargement [18]. The authors further demonstrated that bladder obstruction causes microscopic ganglial hypertrophy, leading to denervation super sensitivity and abnormal afferent and efferent signals as part of a “denervation and reinnervation” pattern [18,19]. C-fiber-mediated spinal reflex patterns are also altered as a sequence of a signaling disturbance enhancing detrusor oversensitivity [18,19,20]. 

Bladder oversensitivity, as defined by the International Continence Society (ICS), is an increased perceived bladder sensation during bladder filling with cystometric findings of early desire to void without an increase in detrusor pressure [8] and is considered according to the work of Yamaguchi et al. as part of the OAB characteristics [21].

Our results imply that perhaps bladder oversensitivity as a characteristic OAB symptom is a preliminary clinical and urodynamic finding prior to overt DO during the detrusor remodeling phase and the gradual development of full-spectrum OAB symptoms. 

One can hypothesize that early administration of alpha blockers medication to patients with bladder hypersensitivity will perhaps reduce or even prevent overt OAB symptoms. 

There were a few limitations to this study. Firstly, our study was limited due to a relatively small group of non-obstructed men, which can be explained by a referral bias. Secondly, air-charged catheters are not ICS standard; however, all subjects were tested with the same type of catheters, enabling assessment and analysis of the groups. Lastly, despite the accepted bladder filling rate of 50 mL/min, a slower filling rate could have perhaps changed the sensation results. Further validation will be needed to reconfirm our findings in a larger cohort. Changes in sensation volumes in post-obstruction-treatment urodynamic studies is another important question that should also be investigated in the future.

To our knowledge, this is the first prospective study in men demonstrating bladder oversensitivity as a potential neuromodulatory sequence of bladder outlet obstruction. This analysis revealed a probable physiological connection that should be further investigated due to the potential benefit of early intervention in LUTS management and interpretation of bladder oversensitivity. 

## 5. Conclusions

Our results suggest that there is a strong association between bladder oversensitivity and FDV as a marker for BOO in men. This may shed light on the pathophysiological relation between obstruction and enhanced bladder afferent signaling.

## Figures and Tables

**Figure 1 jpm-12-01675-f001:**
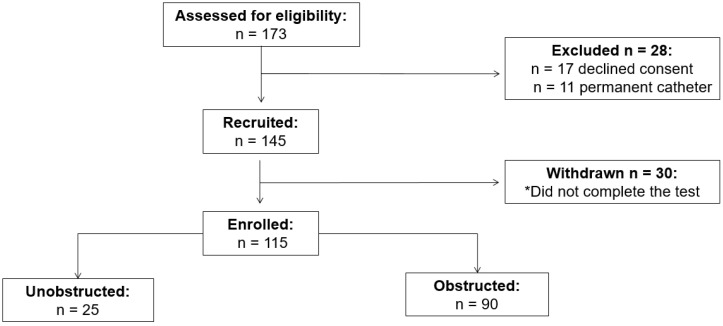
CONSORT diagram—flow of participants through trial phases. * Due to inability to insert the urethral probe or inability to void with the urethral catheter in place.

**Figure 2 jpm-12-01675-f002:**
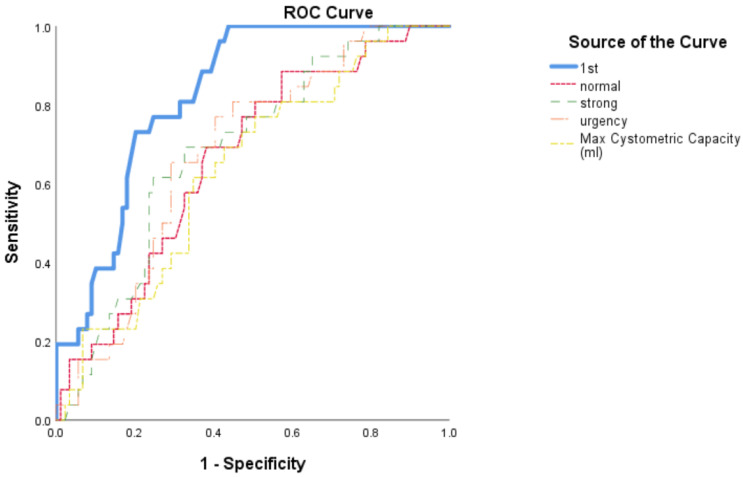
ROC curves of all sensation parameters. First desire to void (1st) with AUC = 0.83, 95% CI = 0.76–0.90, *p* < 0.001. The Youden index in the ROC curve defines a cutoff of 0.78 at 140 mL for predicting a threshold to outflow obstruction with sensitivity of 83% and specificity of 81%.

**Table 1 jpm-12-01675-t001:** Comparison of demographics, clinical and urodynamics variables between obstructed and unobstructed men.

*p*-Value	Unobstructed *n* = 25	Obstructed *n* = 90	
0.25	54 ± 12	57 ± 13	Age, mean ± SD
0.17	29 ± 5.9	30 ± 6.3	BMI, mean ± SD
			Comorbidities:
0.14	8 (32)	25 (27)	NIDDM, *n* (%)
0.67	9 (36)	27 (30)	Hypertension, *n* (%)
0.41	3 (12)	9 (10)	Ischemic heart disease, *n* (%)
0.43	2 (8)	8 (9)	Hypothyroidism, *n* (%)
0.11	10 (40)	31 (34)	Hypercholesterolemia *n* (%)
0.51	0 (0)	1 (1)	Parkinson’s, *n* (%)
0.55	3 (4)	0 (0)	Previous CVA, *n* (%)
0.003	0.45 (0.05–1.11)	1.48 (0.7–3.17)	PSA, median (IQR)
			Previous surgeries:
0.51	0 (0)	1 (1)	Urinary incontinence surgery, *n* (%)
			Medication:
0.66	4 (16)	14 (15)	Alpha receptor blockers, *n* (%)
0.52	1 (4)	3 (3)	Antimuscarinic, *n* (%)
			IPSS:
0.01	17 (13–24)	22.5 (16–29)	Total score, median (IQR)
0.1	4 (3–5)	5 (3–5)	QOL score, median (IQR)
			Physical examination:
0.12	4 (16)	12 (13)	Enlarged prostate, *n* (%)
0.24	0 (0)	2 (2)	Meatal stenosis, *n* (%)
			Ultrasonography findings:
0.51	1 (0)	1 (1)	Urolithiasis, *n* (%)
1	0 (0)	0 (0)	Hydronephrosis, *n* (%)
0.41	1 (4)	6 (7)	Trabeculated bladder, *n* (%)
0.04	30 (20–35)	35 (20–50)	Estimated prostate size (cc), median (IQR)
0.38	38 (10–51)	47 (15–68)	Estimated PVR (mL), median (IQR)
			Urodynamic parameters:
0.31	52 (12–60)	63 (15–79)	PVR (mL), median (IQR)
<0.0001	14 (9–20)	8.1 (4–12)	Qmax @ free flow (mL/s), median (IQR)
0.01	6.8 (3–10.8)	4.5 (2–7)	Qavg @ free flow (mL/s), median (IQR)
0.76	11 (44)	40 (44)	No. detrusor overactivity (%)
0.58	3 (11)	14 (15)	No. low compliance (%)
0.11	7 (28)	11 (12)	No. UUI (%)
0.08	3 (12)	0 (0)	No. SUI (%)
<0.0001	180 (80–200)	118 (57–128)	First desire, median (IQR)
0.008	227 (125–350)	171 (85–257)	Normal desire, median (IQR)
0.047	307 (180–477)	221 (150–383)	Strong desire, median (IQR)
0.045	344 (280–535)	276 (197–480)	Urgency, median (IQR)
0.26	355 (345–490)	390 (250–500)	MCC, median (IQR)

BMI—body mass index; NIDDM -non-insulin-dependent diabetes mellites; CVA—cerebrovascular accident; PVR—post void residual; OAB—overactive bladder; IPSS—International Prostate Symptoms Score; UUI—urge urinary incontinence; SUI—stress urinary incontinence; DO—detrusor overactivity; MCC—maximal cystometric capacity.

**Table 2 jpm-12-01675-t002:** Association between bladder sensation volumes and BOO according to univariate and multivariate logistic regression analysis.

	Univariate OR (95% CI, *p*-Value)	Multivariable * OR (95% CI, *p*-Value)
First desire to void (50 mL ↑)	1.46 (1.17–1.81, *p* = 0.0001)	1.42 (1.25–1.68, *p* = 0.001)
Normal desire. Median (50 mL ↑) (IQR)	1.12 (1.01–1.22, *p* = 0.04)	1.05 (0.96–1.22, *p* = 0.08)
Strong desire. median (50 mL ↑)	1.18 (1.04–1.22, *p* = 0.05)	1.14 (1.01–1.29, *p* = 0.09)
Urgency. Median (50 mL ↑) (IQR)	1.22 (1.05–1.36, *p* = 0.004)	1.04 (0.98–1.22, *p* = 0.06)
MCC (50 mL ↑)	1.01 (0.97–1.12, *p* = 0.07)	1.01 (0.96–1.22, *p* = 0.12)

MCC = maximal cystometric capacity; OR = odds ratio; CI = confident interval; IPSS = International Prostate Symptoms Score; USI = urodynamic stress incontinence. 50 mL ↑ = adjusted to potential decline of 50 mL. * Adjusted to previous CVA, PSA, TURP, IPSS, and estimated prostate size.

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
