# Peer review of "Bladder Oversensitivity Is Associated with Bladder Outlet Obstruction in Men"

_jpm, 2022, doi:10.3390/jpm12101675_

Round 1
Reviewer 1 Report
This study aims to assess whether there is an objective association between bladder outlet obstruction (BOO) and abnormal sensation parameters during bladder filling. They concluded that there is a strong association between bladder oversensitivity and BOO in men, in fact, men with FDV<140 mL had a significantly increased probability of being obstructed. I believe that the study has sufficient merit to be considered for publication in Journal of Personalized Medicine, although major revisions are required.
MAJOR COMMENTS
- In line 29 specify the unit of measures (FDV 140).
- Keywords: the acronym BOO is unnecessary.
- Introduction. In lines 44-45 the authors should better discuss the reason for bladder overactivity post-prostatectomy (the exact mechanisms underlying the relationship between the OAB symptoms and radical prostatectomy are not fully elucidated). I suggest providing a more detailed presentation of bladder outlet obstruction and morpho-functional alterations involving the detrusor muscle; in this regard, discuss this interesting paper (doi: 10.4081/aiua.2020.3.173; PMID 33016038).
- In exclusion criteria, the authors should underline the possible presence of urethral stenosis, diabetes not well compensated, or chronic renal failure.
- Discussion. The authors should argue more about the discussion, including what may be the role of therapy in these patients.
Author Response
Dear Reviewer,
First and foremost we would like to thank you for the time you dedicated to revise our manuscript. It is GREATLY appreciated. Below you will find our responses to your comments. You will find a detailed explanations to the points of controversy and we do hope that we managed to convince you. We also hope, that following all the corrections you will find our paper suitable for acceptance to the Journal of Personalized Medicine.
Thank you for all your suggestions, including the article reference! We have followed them and corrected accordingly. We marked the possible role of early BOO therapy in the treatment of oversensitivity and the potential therapeutic effect of alpha blockers on early OAB symptoms including urge and oversensitivity symptoms.
Reviewer 2 Report
Dear authors,
First, congratulate them for their findings, they will undoubtedly contribute to their research field.
Here are some suggestions;
Line 41, Add reference "... 57% Overlap."
Line 48, Add reference "... perception"
Line 52 and 54, define whether the reference number will be attached to the text or leave a space between the text and the number. Check throughout the manuscript.
Line 58, individualize the Ethics Committee that the study approved.
Line 67, add reference for the IPSS questionnaire, I suggest adding the questionnaire used as supplementary material.
Line 120, in the table in the "Medication" section, 2 types of drugs are indicated, but in line 70 they indicate 3 types of drugs that patients could be using. Please clarify this point.
Line 145, I suggest adding the graphic legend (Figure 2) under the curve or present in another way.
I hope these suggestions are welcomed for the best understanding of your findings.
Author Response
Dear Reviewer,
First and foremost we would like to thank you for the time you dedicated to revise our manuscript. It is GREATLY appreciated. Below you will find our responses to your comments. You will find a detailed explanations to the points of controversy and we do hope that we managed to convince you. We also hope, that following all the corrections you will find our paper suitable for acceptance to the Journal of Personalized Medicine.
Thank you for all your suggestions! We have followed them and corrected accordingly.
Regarding the Medications section, none of the participants used beta 3 agonists.
Reviewer 3 Report
The authors conducted the same study on women two years ago and published it in Int Urogynecol J. 2021;32(10):2771-2776, i.e. Ref. 9. The contribution of this work is showing the same conclusion on men. So, the novelty of this study is a big concern. In addition, the language needs excessive editing.
Author Response
First and foremost we would like to thank you for the time you dedicated to revise our manuscript. It is GREATLY appreciated. Below you will find our responses to your comments. You will find detailed explanations to the points of controversy and we do hope that we managed to convince you. We also hope, that following all the corrections you will find our paper suitable for acceptance to the Journal of Personalized Medicine.
- According to your suggestion, the paper was sent to an additional English language editing.
- Our previous findings as were published before did find an association between bladder oversensitivity and bladder outlet obstruction in women. As the etiology of the obstruction is different from those of men including the pressure changes in the bladder needed to void in both genders, as well as that the possible pathophysiologic changes to the bladder may differ, we believe that validating a similar trend in men is an important step for future studies looking at the clinical implementations of these findings.
Round 2
Reviewer 1 Report
I believe that the study has sufficient merit to be considered for publication although minor revisions are required.
The corrections made have certainly improved the work. However, I would like to develop these two points as reported in the first review , reporting the right references :
Introduction. In lines 44-45 the authors should better discuss the reason for bladder overactivity post-prostatectomy (the exact mechanisms underlying the relationship between the OAB symptoms and radical prostatectomy are not fully elucidated). I suggest providing a more detailed presentation of bladder outlet obstruction and morpho-functional alterations involving the detrusor muscle; in this regard, discuss this interesting paper (doi: 10.4081/aiua.2020.3.173; PMID 33016038).
Author Response
Dear Reviewer,
First and foremost we would like to thank you again for the time you dedicated to revise our manuscript. It is GREATLY appreciated. Below you will find our response to your comment. We do hope that we managed to convince you. We also hope, that following all the corrections you will find our paper suitable for acceptance to the Journal of Personalized Medicine.
The corrections made have certainly improved the work. However, I would like to develop these two points as reported in the first review , reporting the right references :
Introduction. In lines 44-45 the authors should better discuss the reason for bladder overactivity post-prostatectomy (the exact mechanisms underlying the relationship between the OAB symptoms and radical prostatectomy are not fully elucidated). I suggest providing a more detailed presentation of bladder outlet obstruction and morpho-functional alterations involving the detrusor muscle; in this regard, discuss this interesting paper (doi: 10.4081/aiua.2020.3.173; PMID 33016038).
Thank you so much for your suggestion regarding the article. We found it very interesting, with a wide and detailed presentation of the possible mechanism underlying the relief of OAB symptoms after prostatectomy. We included the mechanism in the text.
Reviewer 3 Report
I persist with my opinion. I believe the studies for men and women should be put together in one single report and make it comprehensive instead of divided into two without any new insights. If your way can stand, all the data can be reported in two separate papers, men vs. women, even with the same conclusion. In the current manuscript, maybe a comprehensive comparison and discussion about the difference and new adding compared to the previous studies can help. Nevertheless, I don't think I will change my opinion.
Author Response
Dear Reviewer,
First and foremost we would like to thank you again for the time you dedicated to revise our manuscript. It is GREATLY appreciated. We hope, that following our response you will find our paper suitable for acceptance to the Journal of Personalized Medicine.
I persist with my opinion. I believe the studies for men and women should be put together in one single report and make it comprehensive instead of divided into two without any new insights. If your way can stand, all the data can be reported in two separate papers, men vs. women, even with the same conclusion. In the current manuscript, maybe a comprehensive comparison and discussion about the difference and new adding compared to the previous studies can help. Nevertheless, I don't think I will change my opinion.
- There is a basic physiologic difference between male and female bladder function regarding storage and voiding phase, mainly due to pressure difference, urethral length, supporting structures influenced by hormone (estrogen) difference and bladder remodeling. There is also difference in evaluated parameters such as menopause and labor history in women and PSA and prostate size in men. Our previous findings as were published before did find an association between bladder oversensitivity and bladder outlet obstruction in women, and as a consequence of that data we gathered new data regarding men to examine the pattern despite the described difference. We believe that in this particular case there is physiologic interest to describe the phenomena separately in the different genders.